# Local Breast Microbiota: A “New” Player on the Block

**DOI:** 10.3390/cancers14153811

**Published:** 2022-08-05

**Authors:** Marina Vitorino, Diogo Alpuim Costa, Rodrigo Vicente, Telma Caleça, Catarina Santos

**Affiliations:** 1Medical Oncology Department, Hospital Professor Doutor Fernando Fonseca, 2720-276 Amadora, Portugal; 2Breast Cancer Unit, CUF Oncologia, 1998-018 Lisbon, Portugal; 3NOVA Medical School, Faculdade de Ciências Médicas, 1169-056 Lisbon, Portugal

**Keywords:** microbiota, microbiome, cancer, breast cancer, immune system, immune microenvironment, treatment, immunotherapy

## Abstract

**Simple Summary:**

Microbiota plays a fundamental role in the induction, training and function of the human immune system. The interactions between microbiota and immune cells have consequences in several settings, namely in carcinogenesis but also in anticancer activity. Immunotherapy, already widely used in the treatment of several solid cancers, modulates the action of the immune system, promoting antitumour effects. Recently, there has been a growing interest in studying the microbiota composition as a possible modulator of the tumour microenvironment and consequently of the response to certain therapies such as immunotherapy.

**Abstract:**

The tumour microenvironment (TME) comprises a complex ecosystem of different cell types, including immune cells, cells of the vasculature and lymphatic system, cancer-associated fibroblasts, pericytes, and adipocytes. Cancer proliferation, invasion, metastasis, drug resistance and immune escape are all influenced by the dynamic interaction between cancer cells and TME. Microbes, such as bacteria, fungi, viruses, archaea and protists, found within tumour tissues, constitute the intratumour microbiota, which is tumour type-specific and distinct among patients with different clinical outcomes. Growing evidence reveals a significant relevance of local microbiota in the colon, liver, breast, lung, oral cavity and pancreas carcinogenesis. Moreover, there is a growing interest in the tumour immune microenvironment (TIME) pointed out in several cross-sectional studies on the correlation between microbiota and TME. It is now known that microorganisms have the capacity to change the density and function of anticancer and suppressive immune cells, enabling the promotion of an inflammatory environment. As immunotherapy (such as immune checkpoint inhibitors) is becoming a promising therapy using TIME as a therapeutic target, the analysis and comprehension of local microbiota and its modulating strategies can help improve cancer treatments.

## 1. Introduction

The “microbiota” refers to the set composed of resident microbes on and inside the body [1]. Different microbiota ecosystems in the human body, such as the gastrointestinal tract, skin, vaginal mucosa or oral cavity, account for trillions of microorganisms [2]. New evidence documented that the microbiota influences the oncogenesis process and anticancer treatment outcomes by regulating local and systemic antitumour immunity [3,4]. The tumour microenvironment (TME), and more specifically, the tumour immune microenvironment (TIME), can promote cancer progression or prevent the growth of malignant cells, depending on the type of cells and the signals of TME [5]. The microbiota, especially those adjacent to tumour cells, can influence the interactions between the TME and the tumour. Antitumour immune activity can be stimulated or inhibited through signalling pathways, which in turn can be composed of microbe-derived polysaccharides [6]. This association has been described and characterised in more recent studies, highlighting the importance that variations in the microbiota can lead to more or less favourable tumour responses [1]. Based on this association between host microbiota and immune response, it is suggested that the manipulation of microbiota constitution may provide an adjuvant strategy to anti-neoplastic therapies, namely with the use of the immune checkpoints inhibitors (ICI) [7].

In this review, we discuss concepts related to microbiota, TIME and carcinogenesis, the importance of local microbiota in different types of cancer, mainly in breast cancer (BC) and the future therapeutic implications.

## 2. Human Microbiota and the Relation with the Host

Near 100 trillion dynamic microorganisms from 5000 different species, including bacteria, viruses, fungi, archaea and protists, inhabit the human body in different locations, such as the gastrointestinal tract, skin, vaginal mucosa or the oral cavity and play different roles in immune system regulation, inflammatory state, tolerance for commensal bacteria, recognition of potentially infectious pathogenic organisms, intestinal permeability, energy balance and endocrine hormone secretion. The “microbiota” refers to the set composed of resident microbes on and inside the body, and the “microbiome” is the collective genome of these biological agents [1,8].

Human microbiota composition is distinctive to each individual, probably starting before birth. There is growing evidence that placenta, amniotic fluid and meconium microbial flora include non-pathogenic commensal microbes, which probably contributes to a possible heritage of maternal microbiota and foetal immune system development [9,10,11].

Acquisition of significant amounts of microbiota occurs during and immediately after birth and develops during the first three or four years of life, influenced by breastfed, household exposures, chronic conditions and geographic location. After that period, microbiota composition becomes relatively stable, only slightly modified throughout adulthood by host genetics, diet, lifestyle and diseases [9,10].

Regarding microbiota, its complexity can be described using the concepts of *alpha*-diversity, that describes the richness in a given sample (i.e., number of organisms and distribution of those organisms), and *beta*-diversity, that defines the extent of relative or absolute overlap of a microbial community between different samples [12].

Resilience is related to microbiota capacity for self-regeneration and restoration of homeostasis after any shift in its composition. However, in some cases, the microbiota cannot remain resilient after a perturbation, leading to a new equilibrium state, called “dysbiosis”. Dysbiosis, an altered composition of commensal microbiota and its metabolic activity, causes an imbalance in the symbiosis between the host and its organic habitat. Therefore, this deregulation can harm the human host and influence the onset of various inflammatory, auto-immune or malignant conditions [4,9].

For that reason, human resident microbiota and its complex relation with the host are now emerging as important elements in the lifelong maintenance of health and immune system homeostasis, with substantial attention given to its influence on cancer cell proliferation, tumourigenesis, disease progression and treatment outcomes [3,4].

Despite centuries of historical reports linking cancer and microbes, the International Agency for Research on Cancer (IACR) just considers 11 of the ~10^12^ microbial species on earth to directly cause cancer. However, it is suggested that approximately 20% of human cancers may be linked to microbial pathogens [12]. Several oncogenic microbes drive cancer, with *Helicobacter-pylori*-induced gastritis and gastric adenocarcinoma being perhaps the best evidence that the microbiota is not just a bystander in the cancer development process [13]. *H. pylori.* infection can contribute to the release of virulence factors that cause cellular stress in gastric epithelium, affecting host cell signalling pathways. Eradication of *H. pylori.* is an important method of reducing the risk of gastric cancer [14]. Previous studies have also established a causal link between the gut bacterium *Bacteroides fragilis*, oral pathogen *Fusobacterium nucleatum*, *Escherichia coli* and colorectal cancer [15,16,17,18]. The presence of *F. nucleatum* is associated with malignant transformation of colorectal adenomas to carcinoma and is also related to a worse survival of colorectal patients [14]. Metagenomic sequencing studies have detected significant differences in the composition of microbial communities in numerous human cancers compared to controls with normal tissues [12,19,20]. Altered bacterial diversity in faecal samples was also associated with colorectal cancer by Ahn et al. [19]. Samples were found to be depleted of Firmicutes, with a lower abundance of *Clostridium* and higher levels of the pro-inflammatory genera *Fusobacterium* and *Porphyromonas* [20].

The diversity of microbial populations and the host’s physiologic environments at different human body sites suggests that microbial mechanisms and species involved in cancer onset will also vary depending on the location. It has also been established that tumours are non-sterile environments harbouring bacteria. In a recent extensive study, tissue from more than 1500 samples regarding seven human tumour types, including breast, lung, melanoma, ovary, brain, bone and pancreas, were examined and compared using a bacterial 16S rDNA PCR sequencing technique. Nejman et al. deduced that each tumour type has a distinct (local) microbiota composition and that BC has a particularly rich and diverse microbiota and also confirmed the presence of cytoplasmatic bacteria in both tumours and immune cells [21].

Although not fully clarified, various mechanisms of dysbiosis-induced cancer have been proposed in several studies: induction of inflammatory microenvironment and epithelial–mesenchymal transition (EMT), increase in reactive oxygen species (ROS) and DNA damage, genotoxic substances gathering, suppression of antitumour immune response and destruction of the gut mucosal layer with changes in intestinal permeability that allows translocation of pathogens and its byproducts to surrounding tissues and systemic circulation [1,22,23,24,25,26].

Although microbiota influences carcinogenesis through mechanisms independent of inflammation and immune system, the most recognised link is between microbiota and cancer via its effects on innate and adaptive immunity, modulating both local and systemic immune responses of the host [1,2,27]. This association is particularly strong between the gut microbiota and intestinal mucosal immune system. Pattern recognition receptors (PRRs), like Toll-like receptors (TLRs), are expressed in the human body by many cells, including immune cells, and act as detectors of pathogen components. Through microbe- or pathogen-associated molecular patterns (MAMPs or PAMPs), microbes interact with these receptors, activating inflammatory pathways and causing a cytokine release [1,2,27].

In addition, bacterial metabolites and byproducts also directly interfere with immune local cells’ actions, stimulating the maturation of local dendritic cells (DCs) through interaction with PRRs. These cells travel from their area to mesenteric lymph nodes, triggering lymphocyte differentiation of naïve CD4+ T cells into regulatory T-lymphocytes (Tregs) and T helper 17 (Th17). After maturation, effectors T cells can travel back to their original place to regulate local immune responses while another subset migrates to systemic circulation and influences immunity in different sites. For example, circulating Th17 cells enhance antitumour immunity, protecting against bacterial and fungal infections and circulating Tregs secrete anti-inflammatory cytokines. Another example of a direct link between microbiota and local immune response is its impact on B cells as the main mediator of gut mucosal homeostasis through the production of immunoglobulin A, which blocks bacterial adherence to epithelial cells [1,2,27].

Environmental factors such as inappropriate diet patterns are also important contributors to alterations in microbiota diversity. Microbes use ingested nutrients for harvesting energy and basic biological processes. Consumption of high levels of red meat is a risk factor for colorectal cancer and several other cancers by various mechanisms, some of them dependent on gut bacteria. Increased colonic protein levels intake can lead to increased bacterial fermentation of amino acids to N-nitroso compounds that induce DNA alkylation and mutations in the host. High fibre, low-fat diets are also capable of shifting the microbiota community towards the advantageous bacteria and increasing microbiota-derived short-chain fatty acids (SCFAs), like butyrate, a pleiotropic molecule that exerts its tumour-suppressive properties by multiple mechanisms and has been implicated in colorectal cancer prevention based on metagenomic studies and mouse models [1,12].

Faecal microbiota transplant (FMT) is an emerging therapeutic approach in many potential applications and has primarily been applied in patients with relapsed/refractory *Clostridioides difficile* infection. Due to the complexity of the diseases and their treatment, patients with haematologic and oncologic diseases are particularly susceptible to complications related to altered intestinal microbiota [28]. Currently, there are nearly 40 studies registered that primarily evaluate the safety of FMT, the use of FMT following allogeneic hematopoietic stem cell transplantation, improvement in ICI response, and the treatment of the complications that arise due to cancer therapy [28]. Various retrospective studies suggested a possible relation between broad-spectrum antibiotics, altered intestinal microbiota and its negative impact on responses to ICI treatment in cancer patients [29,30,31,32]. Based on these findings, two studies were performed, aiming to determine the safety and feasibility of FMT before re-introducing immunotherapy in refractory malignant melanoma. This treatment increased the intratumour immune activity in some patients, translated into objective clinical responses. These results support the concept of overcoming resistance to immunotherapy by modulating gut microbiota [33,34].

## 3. Tumour Microenvironment

The tumour microenvironment (TME) relates to cancer cells and all types of cells surrounding them, including immune cells, blood vessels, extracellular matrix, fibroblasts, lymphocytes, signalling molecules such as cytokines, growth factors and enzymes (Figure 1) [35,36]. Interactions between these two types of cells, malignant and non-malignant, affect the tumour, the process of carcinogenesis, proliferation of malignant cells, and progression of the tumour. These interactions contribute to the host’s tolerance and response to the tumour. The mechanisms that allow tumour proliferation include angiogenesis, inhibition of apoptosis, immune system suppression, and are all controlled by cells of TME [5]. Growing evidence of this relationship between cancer and TME also increases the interest in TME as a prognostic factor and a potential therapeutic target [36].

### 3.1. Tumour Immune Microenvironment

The host immune system is an important key to regulating cancer development. The recruitment, activation and response of immune cells will determine the capacity of tumour cell proliferation. The comprehension of density and phenotype of immune cells and the cytokines secreted can have prognostic implications and influence therapeutics’ success [35,37]. The TIME can promote cancer progression. However, it can likewise prevent the growth of malignant cells, depending on the type of cells and the signals of TME, whether pro- or anti-inflammatory, respectively [5,38].

Tumour-infiltrating lymphocytes (TILs) are key cells in the TIME, with the majority being T cells. According to their T cell receptor (TCR), T cells are classified into CD4+ Th cells, Tregs, CD8+ T cells and natural killer cells (NK) [36]. Some T cells are related to tumourigenesis, like Treg or helper T cells, whereas others contribute to eliminating the tumour, like NK and cytotoxic T cells [35]. Treg cells block the immune response by expressing cytokines against antitumour cells. The secretion of interleukin-10 (IL-10) or transforming growth factor-*beta* (TGF-β) can suppress effector cells, such as CD8+ cells, creating an immunosuppressive microenvironment. The tumour itself can recruit Treg cells through the secretion of certain signals, such as prostaglandin E2 (PGE2) and TGF-β. High numbers of Treg cells are associated with a worse prognosis in various types of cancer, including pancreatic, ovarian and BC. On the other hand, infiltration by CD8+ cells is associated with better cancer-related outcomes. CD8+ T cells have the capacity to produce high levels of antitumour cytokines such as tumour necrosis factor-alpha (TNF-α) or interferon-gamma (IFN-γ) [39]. This antitumour activity of CD8+ cells is regulated by the balance between co-stimulatory and co-inhibitory signals, called immune checkpoints. The inhibitory stimulus influences T cells´ function and the other cells of TIME, creating an immunosuppressive environment. Anti-T-lymphocyte-associated antigen 4 (CTLA-4), anti-programmed death 1 (PD-1) or its ligand PD-L1 are ICI that revolutionised the use of immunotherapy as an emergent weapon in cancer therapies. Targeting the inhibitory signals with antibodies makes it possible to eliminate the antitumour activity and consequently reduce the tumour progression [36].

NK cells are characterised by the ability to rapidly mediate cytotoxicity [39]. These cells can be recruited to sites of tumour growth through pro-inflammatory cytokines produced by other immune cells of the TIME. In parallel, NK cells can also affect the TIME by self-production of cytokines that modulate the immune response. Similarly to other TIME constituents, NK cells are also suppressed by inhibitory signals produced by tumour cells [40]. Hypoxia, which can be present in some solid tumours, often turns into a barrier to NK cell’s action, downregulating receptors and modifying the cytokine secretion [41].

B cells are more predominant in margin tissues surrounding the tumour. Like other cells, B cells can have a pro-tumour or an antitumour activity according to the constitution and signals of the TME [42]. In addition, B cells can modulate the immune response through antibody production and increase the T cell activation with antigen presentation. However, B cells can also have an inhibitory behaviour, suppressing the effector T cells and NK cells and expanding Treg cells [43].

DC present tumour antigens to T cells, CD4+ or CD8+, which can induce their response to eliminate malignant cells. However, this capacity of DC is dependent on their grade of maturation. Immature DC produce proangiogenic factors, promoting angiogenesis and tumour growth. A more intense infiltration of immature DC in analysis of rapid growth tumours has been reported, associated with more aggressive neoplasms [36,44]. DC metabolism is modulated by TME, by secreted cytokines and upregulation of transcriptional and metabolic pathways that promote a tolerogenic environment, where there is a prevalence of immunosuppressive factors such as vascular endothelial growth factor (VEGF), IL-10, TGF-β, PGE2 and other cytokines [45]. Manipulation of DC and their maturation is an interesting point for developing new therapies capable of inducing antitumour immune responses in cancer patients [46].

One of the most important cells in TIME are the macrophages, called tumour-associated macrophages (TAM). This cell population is strongly related to tumour progression because of its ability to improve angiogenesis and suppress antitumour immunity. Theoretically, there are two types of macrophages, the M1 phenotype, with more pro-inflammatory activity, and the M2 phenotype, which is predominant in TIME and has more anti-inflammatory properties [36]. M2 macrophages promote the escape of tumour cells into the circulatory system and can suppress antitumour immune mechanisms and responses [35]. Through secretion of some cytokines, such as IL-10 and TGF-β, macrophages allow cancer cells to survive and disseminate. M2 macrophages are also responsible for the recruitment of cells that inhibit the antitumour activity, such as CD4+ Tregs, in place of cytotoxic cells [37]. Furthermore, TAM can facilitate the invasion of the surrounding tissue by matrix deposition and remodelling, leading to metastasis formation [36]. High infiltration of TAM in cancer tissues is correlated, in different malignancies, to a worse prognosis [45].

By understanding the characteristics of the TME and targeting specific components, such as factors promoting angiogenesis, it is expected that cancer growth and metastasis will be further inhibited and a lasting therapeutic effect will be achieved [47]. Antiangiogenic therapy normalises tumour vasculature, improving local perfusion, relieves TME hypoxia and reverses the immunosuppressive state, promoting the aggregation of tumour-infiltrating immune cells in TME, which forms the basis for the synergistic relationship between antiangiogenic therapy and immunotherapy [48]. Several pivotal clinical trials have already demonstrated the superiority of combining antiangiogenic agents and ICIs in various malignancies, such as non-small cell lung cancer, hepatic cell carcinoma, and renal cell carcinoma. However, unsatisfactory results in some tumours, such as BC, make this approach an ongoing challenge, with several unanswered questions, such as overcoming the resistance to anti-angiogenesis agents or identification of biomarkers [47].

The constitution of the TIME has important prognostic value and can be a histopathological and molecular biomarker in evaluating patient responses to treatment [35].

### 3.2. Tumour Microbe Microenvironment

Recent studies have allowed the identification and characterisation of the tumour type-specific microbiota [49]. Nejman et al. demonstrated that different tumours have different microbiota compositions, which can be correlated with clinical characteristics, tumour behaviour and response to therapies [21]. Microbes can originate from the tumour tissue or migrate from distant organs or metastasis [50]. In addition, the gut microbiota can translocate via circulation and lodge in the tumour bed. The analysis of the constitution of tumour microbiota is complex, and the methods used differ in various studies. This heterogeneity of methodology also contributes to some confounders such as diet, medications or geographical location [51].

The tumour microbe microenvironment composed of bacteria, fungi, viruses or mycoplasma. These agents and their metabolites can interact with tumour cells and immune cells in the TIME, contributing to modifications of their actions [50]. The role of viruses in tumours is relatively more well established [52,53]. This review will focus on the evidence about bacteria and their potential in tumours.

Previous studies showed that the response to oncological therapies, such as immunotherapy, can be related to the predominance of certain bacteria. *Fusobacteria* or *E. coli* are associated with better outcomes in colon cancer patients treated with ICI [54]. Similar studies with other types of cancer, such as breast, gastric or lung cancer, demonstrated the interaction between microbes and their metabolites with the TIME [18,55,56].

There are some mechanisms hypothesised to explain the relationship between microbes and cells of TIME [50]. One of them is the potential of microbe antigens to mimic tumour antigens. The microbes’ antigens can be present to immune cells and the tumour cells, and this presentation can trigger an immune response with recognition and killing of the antigen-presenting cell by effector T cells [57]. Due to antigen mimicry, T cells can recognise tumour cells that present similar antigen epitopes and, consequently, kill them [58]. Another important mechanism is the microbes’ capacity to modulate the TIME through the interaction with PRR, like TLR. Microbes can interact with different PRRs, with stimulatory or inhibitory finality. The sum of these interactions with different PRRs will drive a final response through signalling pathways [59]. In the TME, some microbes´ metabolites, such as SCFA or bile acids, are also present. On the surface of tumour cells, there are receptors of some of these metabolites, suggesting that they can also be potential regulators of the functioning of TME [50]. As mentioned above, the immune checkpoints are important keys to regulating the immune cells´ function in the TME. Recent studies observed an interaction between specific bacteria and some ICI. Chauvin et al. suggest that *Fusobacteria* can interact with T cell immunoreceptors through Ig and immunoreceptor tyrosine-based inhibition motif (ITIM) domains (TIGIT), which decrease the activity of immune cells such as NK and cytotoxic T cells [60]. Other similar studies describe interactions between bacteria and immune checkpoints with consequent TIME modifications [26,61].

The intratumour microbiota is different according to the type and subtype of cancer. The proportion of tumours that are positive for bacterial genomic material differs according to the type of tumour; for example, in melanoma it is approximately 15%, and in BC it can reach 60% [21]. In the TME, the bacteria are predominantly founded in the intracellular milieu, cancer cells and immune cells [21].

A comparison between cancer tissue and normal tissues has already been performed. Significant differences in the number and type of bacteria were reported among normal and pathological paired tissues (Figure 2) [62]. Thompson et al. evaluated the RNA sequence to examine the microbiota in breast tissues and reported distinct bacteria, more *Proteobacteria* in tumour samples and *Actinobacteria* in normal tissues [63]. Additionally, Xuan et al. found that the bacterium *Methylobacterium radiotolerans* is relatively frequent in breast tumour tissue, while the bacterium *Sphingomonas yanoikuyae* is more frequent in normal tissue [62]. In addition, Yazdi et al. reported an increased abundance of *M. radiotolerans* in cancer lymph nodes compared with healthy tissue [64]. Wang et al., in contrast, demonstrated decreased levels of *Methylobacterium* in BC tissues, with an abundance of *Corynebacterium*, *Staphylococcus*, *Actinomyces* and *Propionibacteriaceae* [65]. Parhi et al. reported that *F. nucleatum* DNA is abundant in BC cells and accelerates BC progression and metastatic development. *F. nucleatum* can inhibit antitumour immunity by activating immune-suppression checkpoint receptors, TIGIT and CEACAM1, decreasing the CD4 and CD8 T cells. Studies in mouse models demonstrated an increase in metastasis, mainly lung metastasis, in the presence of *F. nucleatum*. However, more studies are necessary to clarify the pro-metastasis mechanisms [55]. Urbaniak et al. explored BC and healthy tissues´ bacteria populations, showing that bacterial profiles differ. Breast cancer tissue has a higher abundance of *Bacillus, Enterobacteriaceae* and *Staphylococcus*. These authors examined specimens of these families, *E. coli* and *Staphylococcus epidermidis* and reported the ability to induce DNA double-stranded breaks, one of the most detrimental types of DNA damage. *Bacillus cereus* has a different procarcinogenic effect, activating 5*α* steroid hydrogenase that can metabolise steroid hormones progesterone and testosterone [66]. The metabolisation of progesterone to 5*α*-3,20-dione(5*α*P) is higher in breast tumours compared to normal tissue and can induce cancer cell proliferation in vitro [67]. Chan et al. investigated the microbial population of nipple aspirate fluid of healthy and BC survivors. This study demonstrated differences in microbes of healthy patients compared with BC samples with a higher incidence of *Alistipes*. In an analysis of *β*-glucuronidase levels, the authors described higher levels of this enzyme in BC samples compared with healthy ones. *Beta*-glucuronidase is a known procarcinogenic enzyme, reversing the conjugation of glucuronide-conjugated oestrogen, leaving active oestrogen and promoting BC [9,68].

Breast microbiome can also have a protective effect against cancer cells. Some studies identified families of bacteria more abundant in normal tissues and with decreased levels in cancer samples. As mentioned above, Xuan et al. reported a prevalence of *S. yanoikuyae* in normal tissues. The presence of *S. yanoikuyae* is associated with some protective mechanisms. Ligands, expressed by *S. yanoikuyae*, can activate invariant NKT cells (iNKT), acting like protectors against cancer cells. This type of bacteria can also change the oestrogen metabolism and activation of TLR 5-dependent pathways that inhibit the development of BC [62,65]. *Lactococcus* and *Streptococcus* are more abundant in healthy tissues than in BC. Activation of NKT cells is also promoted by *Lactococcus*, resulting in a cellular immunity, helping to prevent cancer development [66]. *Streptococcus* can produce anti-oxidant metabolites that neutralise peroxide and superoxide radicals, preventing DNA damage [69]. Urbaniak et al. also reported a greater abundance of *Prevotella* in healthy tissues. These bacteria can produce SCFA that exerts tumour-suppressive properties [66].

More recent research explored differences in local microbiota between different molecular types of BC (luminal A, luminal B, HER2-positive or triple negative—TNBC) [70]. Banerjee et al., using a genome amplification and a pan-pathogen microarray (PathoChip) method, investigated the microbiota diversity in different BC subtypes. As a result, these investigators identified distinct bacterial signatures associated with each type of BC. These conclusions can provide, in the future, tools to assess prognosis and suggest therapies or interventions in addition to standard therapies [70]. Smith et al. also described a different pattern in microbiome constitution of BC tissues according to the molecular type. TNBC was more abundant in phyla Eucaryarchaeota, Cyanobacteria and Firmicutes [71].

## 4. Therapeutic Implications of the Tumour Microenvironment

The TME is characterised by cellular and molecular heterogeneities, where malignant cells, microbiota and immunity have different functions in cancer development. These interactions are reflected in BC tumourigenicity, resulting in different phenotypes and molecular profiles. Currently, the use of specific drugs targeting enzymes (aromatase inhibitors—AI), cell types (osteoclast inhibitors) or cell populations (immunotherapy) plays an important role in clinical practice. However, there is a long way to go [72]. In the last decades, more targetable specific elements on this complex network were identified, but that does not guarantee therapeutic success, as demonstrated by the failure of antiangiogenic agents in BC [73].

Several conventional chemotherapies used in BC, including anthracyclines, cyclophosphamide, platinum salts and gemcitabine, seem to be a part of the modulation of the TIME by the immunogenic cell death process besides their direct antitumour activity. In this mechanism, calreticulin from dying tumour cells is exposed, leading to the antigen presentation to T cells, triggering a cytotoxic immune response towards the neoplastic tissue [74]. However, chemoresistance signalling pathways involving the microenvironment components are also described, leading to tumour recurrence after chemotherapy [75].

Radiotherapy, a recognised treatment technique in adjuvant and palliative settings, has a known immune effect, promoting cross-priming and T-cell response against remaining tumours [74].

Endocrine therapies, such as selective oestrogen receptor modulators (SERMs) or downregulators and AIs, showed opposite effects on the immune system: on the one hand, SERMs lead to decreased intratumour levels of C-C motif chemokine ligand (CCL) 2 and 5 activations of the immune system against metastatic progression; on the other hand, it induces CD4 T cell polarisation on Th2 phenotype avoiding DC functions and suppressing CD8 T cells’ cytotoxic response. In addition, Tregs (FOXP3 T cells) differentiation may be stimulated by AIs, contributing to a more favourable CD8/forkhead box P3 (FOXP3) ratio [76,77,78,79,80].

Another interesting way to target TME is using bone agents (bisphosphonates and denosumab) to improve quality of life and reduce bone recurrences in adjuvant and palliative settings. While bisphosphonates inhibit osteoclastic bone resorption due to the attachment of hydroxyapatite binding sites on bony surfaces, denosumab is a monoclonal antibody that binds and inhibits RANKL (nuclear factor-kB ligand), an important cytokine in the osteoclasts function [81].

Recent studies suggest an important role of local and distant microbiota. A dysbiotic microbiota is responsible for genetic instability, DNA damage, proliferation and inflammatory response modulation, which leads to the multistage process of malignant progression. In fact, microbiota can influence a drug’s efficacy, interfering with its mechanism of action, antitumour effects and toxicity. Most of the drug’s pharmacokinetics and pharmacodynamics depend on certain enzymes exposure, impacting their absorption and bioavailability [67]. Diarrhoea is a known adverse effect caused by many drugs, and specific metabolites such as SN-38 (an active metabolite of irinotecan) or lapatinib (EGFR/HER2 dual tyrosine kinase inhibitor for HER2+) might be involved in alterations to the gut microbiota. Higher levels of *Proteobacteria*, present in much severe diarrhoea or inflammatory diseases, were also found in lapatinib treatment in rats coincident with the higher incidence of diarrhoea [12,82].

Gut microbiota can regulate, as mentioned before, the immune response, affecting, in this way, the response to ICI [7]. The microbiota has been proposed as one of the factors influencing the response to treatment with ICI, and several studies have analysed the microbial composition of samples from these patients. Matson et al. described a higher prevalence of *Bifidobacterium longum*, *Enterococcus faecalis* and *Collinsella aerofaciens* in melanoma patients treated with ICI that had better outcomes. Gopalakrishnan et al. also reported that, in melanoma patients, higher levels of *Faecalibacterium* are associated with a greater abundance of effector T cells, a better response to PD-1 blockade and better outcomes [14,83]. In lung cancer patients, the prevalence of *Alistipes putredinis*, *B. longum* and *Prevotella copri* in responsive patients who were being treated with PD-1 blockade was also reported [7]. Dysbiosis was found to be prevalent in non-responders to anti-PD-1 treatment, with a possible association with inflammation, the block of T cell differentiation and a reduction in the proportion of microbes such as *Sphingomonas*. Other microbes may have a positive impact on immune response: while oral *Bifidobacterium* was associated with an IFN-γ production by CD8+ tumour-specific T cells, *B. fragilis* seem to have an important role in Th1 cell activation and cross-reactivity to tumour neoantigens and bacterial antigens. Benefits were also found in using immune agents such as TLR4 agonists and CpG-oligodeoxynucleotide (CpG-ODN) in a mouse in vivo model, where a worse response was obtained in the microbial-deficient ones. Neoadjuvant treatment in BC was correlated with 65% increases in *Pseudomonas* spp. At the same time, *P. aeruginosa* at high concentrations inhibited the growth of some BC cell lines, enhancing the activity of doxorubicin with bacterial secretions and metabolites. Other studies pointed to the potential of SERMs to avoid infections by *P. aeruginosa*, blocking the biosynthesis pathway of pyocyanin [84].

Breast tumour immunogenicity depends on the subtype: generally, luminal types are considered the less immunogenic, while TNBC is the most inflamed subtype [85]. Given the poor prognosis associated with TNBC, many trials have been conducted using ICI in early and metastatic settings. Moreover, not all patients seem to respond to ICI. Therefore, many measures of immune activity are under investigation, evolving elements related to the TME multiple cells (PD-L1, stromal tumour infiltrating lymphocytes (TILs), bulk tumour gene expression profiling) or to the tumour cells themselves (tumour mutation burden (TMB), DNA damage repair mutation and somatic mutations) [86].

Currently, the value of PD-L1 and TILs is reflected in trials, but only the use of PD-L1 in clinical practice is well established. In the metastatic setting, the IMpassion 130 trial reported that CD8 cell infiltration was predictive of overall survival benefit with atezolizumab while the Ventana SP142 PD-L1 assay also predicted the benefit for that drug [87]. The benefit of pembrolizumab monotherapy compared to chemotherapy was shown in KEYNOTE-119 when stromal TILs ≥ 5%, but PD-L1 positivity alone was not a sufficient biomarker to select a patient for ICI [88]. Additionally, in KEYNOTE-086, TILs were correlated with response rate [89]. TILs are a promising low-cost biomarker and may have additive prediction for response to ICI in the future. There are some restrictions on measuring immune activity since these tools do not assess the function of TILs (CD8 effector, CD4 Th1/Th2, Tregs) or the other cell types that interfere with immunity (DC, NK, and myeloid-derived suppressor cells) [86].

Li Zhu et al. suggest that early-stage BC may have more immunogenicity than metastatic tumours [90]. For this reason, neoadjuvant regimens incorporating ICI are emerging, with two phase III trials reporting initial outcomes and several phase II trials [87,91,92,93].

Taking advantage of the immune modulation mentioned before, the combination of ICI with chemotherapy has been tested. The phase II TONIC trial evaluated the effect of nivolumab after induction therapy with radiation or conventional chemotherapy in advanced TNBC. This study concluded that either doxorubicin or cisplatin might induce a more favourable TME (measured through gene expression profiles) and increase the response to PD-1 blockade [94]. Immunity biomarkers, including the characterisation of TME and gene expression signatures, will be a crucial piece in the future, allowing better prediction of who will benefit from ICI plus chemotherapy or target drugs (NCT0337724 and NCT03742102) or new combinations under investigation (NCT01042379 and NCT03012100) [86].

On the contrary, therapeutics using ICI are revealing some limitations given their mechanism of action. In fact, the inhibitory immune checkpoint blockade cannot suppress the de novo expression of immune checkpoints in tumour cells. Moreover, these monoclonal antibodies cannot substantially regulate oncogenic signalling pathways in TNBC cells [95]. To overcome that, recent evidence is showing that PD-L1 silencing in a process mediated by microRNA (miRNA) may reduce tumour viability, interfering with many mechanisms: it seems to suppress tumour clonogenicity, arrest the cell cycle, stimulate apoptosis, inhibit tumour migration, upregulate pro-inflammatory cytokines and downregulate anti-inflammatory cytokines [96]. The miRNAs are small, non-protein-coding endogenous RNA molecules that are being discussed as crucial in many physiological activities, such as differentiation, cellular proliferation, development and apoptosis [97]. Thus, dysregulation of miRNA expression may lead to tumour development progression and response failure to therapies. Such cancer-derived miRNAs can also modulate immune responses by creating an immunosuppressive TME while downregulating cancer immunogenicity; thus, protecting cancer cells from immune clearance [9]. A recent systematic review described the new findings concerning the effect of these PD-L1-inhibiting miRNAs on TNBC development and antitumour immune responses [95]. Although miRNA-based gene therapy has not been investigated in phase III clinical trials and further studies are required in the field, new therapeutics related to the delivery of miRNAs for treating TNBC are under discussion to provide specific and safe tumour-suppressive miRNA delivery [98,99].

## 5. Future Perspectives

Despite the proven relationship between the microbiota and the possible interference with the treatment and tumour evolution, there are still some limitations to its use. Several confounding factors can influence the microbiota constitution, such as age, sex or the type of diet. In the majority of the studies, these confounding factors are not considered, and frequently, the sample size is insufficient, making the extrapolation of these results to clinical practice difficult. Another important limitation is that most studies evaluate the microbiota in vitro, in isolation from the other important variables in a living organism. In future studies, these previous limitations should be taken into account. The evaluation and characterisation of microbiota should be standardised across studies, choosing a common and feasible method. Furthermore, the settings should be studied depending on the purpose of the study and compared with a generally healthy control group. The results should be combined and categorised into groups according to certain characteristics which are known to interfere with the study objectives. In addition to the analysis of the gut microbiota, further studies evaluating the constitution of the local microbiota should be carried out since it has already been shown that its constitution is variable according to the neoplasm in question. Breast cancer subtypes, besides being prognostic factors, are also associated with different patterns of local microbiota, and therefore, future studies should evaluate these differences. Regarding the different therapeutic options, their relationship with the local microbiota should also be assessed, trying to identify possible predictive relationships of response as well as prognosis.

## 6. Conclusions

Recent literature documented the presence of varied bacterial populations in certain organs, previously considered “steriles”, like the breast. In addition, some differences between healthy and cancer tissues have been demonstrated. Some bacteria are associated with procarcinogenic mechanisms, whereas others can be related to a protective immune response against cancer cells. This relationship between microbiota and immune system also has implications for therapeutic agents’ efficacy and toxicity. Further studies are necessary to clarify the local microbiota in different types of cancer and its capacity to modify the immune mechanisms involved in oncogenesis.

## Figures and Tables

**Figure 1 cancers-14-03811-f001:**
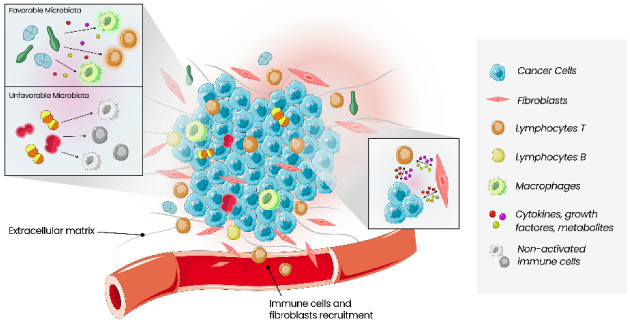
Schematic of the main constituents of the tumour microenvironment. Cancer cells, stromal cell types and immune cells coexist in the tumour microenvironment, interacting via cytokines, growth factors and metabolites. The antitumour activity of immune cells is regulated by the balance between co-stimulatory and co-inhibitory signals. Local microbiota can influence the immune response in a stimulating or inhibitory way, depending on the type of bacteria present.

**Figure 2 cancers-14-03811-f002:**
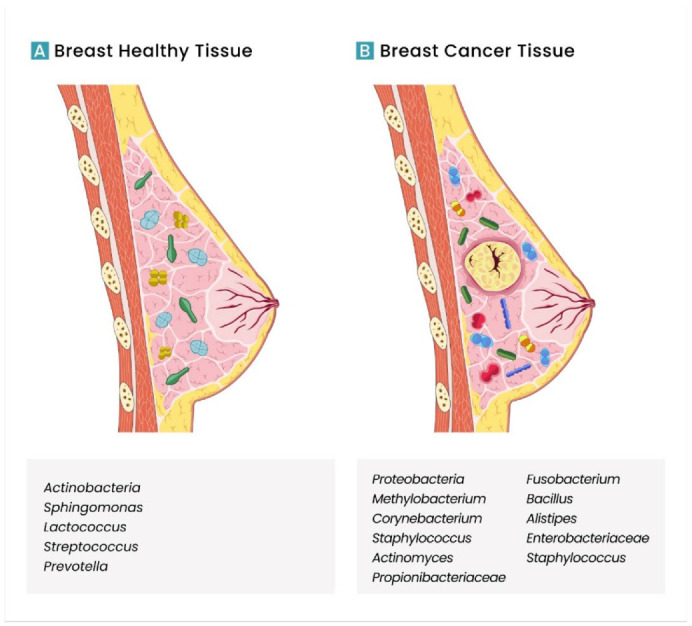
Different microbiota profiles in breast tissues according to different studies. (**A**) Breast healthy tissue; (**B**) breast cancer tissue.

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
