# Peer review of "Local Breast Microbiota: A “New” Player on the Block"

_cancers, 2022, doi:10.3390/cancers14153811_

Round 1
Reviewer 1 Report
Point to be considered:
1) The rationale of why the authors came up with this review.
2) What is the information that is not exactly available that motivated the authors to come up with this information. What are the current caveats and how do the authors highlight the current research in answering them? If not they need to address in future directions.
3) The underlying message here is that more precision and individualized approaches need to be tested in well designed clinical trials – a challenge, but I would be interested in their perspective of how this might be done.
4) As is now well known, tumors grow and evolve through a constant crosstalk with the surrounding microenvironment, and emerging evidence indicates that angiogenesis and immunosuppression frequently occur simultaneously in response to this crosstalk. Accordingly, strategies combining anti-angiogenic therapy and immunotherapy seem to have the potential to tip the balance of the tumor microenvironment and improve treatment response (please expand and comment)
5) In the frame of point 4) thinking, this reviewer personally misses some insights regarding the programmed death-ligand 1 (PD-L1)/programmed cell death protein 1 (PD-1) is a well-established inhibitory immune checkpoint axis in triple-negative breast cancer (TNBC). Growing evidence indicates that tumoral PD-L1 can lead to TNBC development. Although conventional immune checkpoint inhibitors have improved TNBC patients' prognosis, their effect is mainly focused on improving anti-tumoral immune responses without substantially regulating oncogenic signaling pathways in tumoral cells. Moreover, the conventional immune checkpoint inhibitors cannot impede the de novo expression of oncoproteins, like PD-L1, in tumoral cells. Accumulating evidence has indicated that the restoration of specific microRNAs (miRs) can downregulate tumoral PD-L1 and inhibit TNBC development. Since miRs can target multiple mRNAs, miR-based gene therapy can be an appealing approach to inhibit the de novo expression of oncoproteins, like PD-L1, restore anti-tumoral immune responses, and regulate various intracellular singling pathways in TNBC (please refer to PMID: 34440380 and expand the introduction/discussion sections).
6) The authors need to highlight what new information the review is providing to enhance the research in progress
Author Response
Dear Reviewer,
The authors thank the comments and pertinent suggestions. Regarding the suggested topics:
Points 1 and 2: The authors completed the information about the aim of this review in the Introduction
Point 3: The authors considered this topic and completed with information in “Future Perspectives” with some suggestions for future clinical trials.
Point 4: The authors completed the section “3.1 Tumour immune environment” with information about antiangiogenic options
Point 5: The authors completed the section “4. Therapeutic implications of the tumour microenvironment” with information about PD-L1 and TNBC development. Still, in this topic number 4, the authors also completed information about miRNA like it was suggested.
Point 6: the authors consider that throughout the manuscript are described and summarized recent results and theories on the subject in question, microbiota, local microbiota and their implications in the process of tumourigenesis, as well as implications for anti-neoplastic therapy
The changes made in accordance with the comments of the two reviewers are highlighted in the new version of the manuscript.
The authors hope they have been able to fill in the gaps and have added valuable information for a richer manuscript
Reviewer 2 Report
In the current review article, " Local breast microbiota: a “new” player on the block" authors have tried to give a comprehensive outlook on the importance of the local microbiome in breast cancer along with some other forms of cancer and how it can be useful in therapeutic outcomes. Overall, I am very satisfied with the structure of the review. However, I think addressing following comments can help in improving overall impact of the manuscript.
It will be useful if the authors can discuss some studies where fecal matter transplant is investigated in the management of different types of cancers.
Also, the diagrammatic representation showing how the local microbiome influences the TME will be hugely beneficial for the review article.
If authors can discuss in detail the differences in species of bacteria observed across different types of tumors and correlate that to the outcomes of different immunotherapy interventions that will be useful.
Author Response
Dear Reviewer,
The authors thank the comments and pertinent suggestions. Regarding the suggested topics:
- information about fecal transplant has been integrated in section "2. Human microbiota and the relationship with the host"
- the image has been changed according to the suggestions
- the authors describe in more detail the differences in the constitution of the microbiota in the different tumours in section “2. Human microbiota and the relation with the host”. Regarding the outcomes of ICI, information has been supplemented in section "4. Therapeutic implications of the tumour microenvironment
The changes made in accordance with the comments of the two reviewers are highlighted in the new version of the manuscript.
The authors hope they have been able to fill in the gaps and have added valuable information for a richer manuscript
Round 2
Reviewer 1 Report
The authors have clarified several of the questions I raised in my previous review. Most of the major problems have been addressed by this revision.